# Standalone AI Versus AI-Assisted Radiologists in Emergency ICH Detection: A Prospective, Multicenter Diagnostic Accuracy Study

**DOI:** 10.3390/jcm14165700

**Published:** 2025-08-12

**Authors:** Anna N. Khoruzhaya, Polina A. Sakharova, Kirill M. Arzamasov, Elena I. Kremneva, Dmitriy V. Burenchev, Rustam A. Erizhokov, Olga V. Omelyanskaya, Anton V. Vladzymyrskyy, Yuriy A. Vasilev

**Affiliations:** 1State Budget-Funded Health Care Institution of the City of Moscow, Research and Practical Clinical Center for Diagnostics and Telemedicine Technologies of the Moscow Health Care Department, 127051 Moscow, Russia; 2Russian Center of Neurology and Neurosciences, 125367 Moscow, Russia

**Keywords:** intracranial hemorrhage, computed tomography, artificial intelligence, diagnostic accuracy, emergency diagnosis, neuroimaging

## Abstract

**Background/Objectives.** Intracranial hemorrhages (ICHs) require immediate diagnosis for optimal clinical outcomes. Artificial intelligence (AI) is considered a potential solution for optimizing neuroimaging under conditions of radiologist shortage and increasing workload. This study aimed to directly compare diagnostic effectiveness between standalone AI services and AI-assisted radiologists in detecting ICHs on brain CT. **Methods.** A prospective, multicenter comparative study was conducted in 67 medical organizations in Moscow over 15+ months (April 2022–December 2024). We analyzed 3409 brain CT studies containing 1101 ICH cases (32.3%). Three commercial AI services with state registration were compared with radiologist conclusions formulated with access to AI results as auxiliary tools. Statistical analysis included McNemar’s test for paired data and Cohen’s h effect size analysis. **Results.** Radiologists with AI assistance statistically significantly outperformed AI services across all diagnostic metrics (*p* < 0.001): sensitivity 98.91% vs. 95.91%, specificity 99.83% vs. 87.35%, and accuracy 99.53% vs. 90.11%. The radiologists’ diagnostic odds ratio exceeded that of AI by 323-fold. The critical difference was in false-positive rates: 293 cases for AI vs. 4 for radiologists (73-fold increase). Complete complementarity of ICH misses was observed: all 12 cases undetected by radiologists were identified by AI, while all 45 cases missed by AI were diagnosed by radiologists. Agreement between methods was 89.6% (Cohen’s kappa 0.776). **Conclusions.** Radiologists maintain their role as the gold standard in ICH diagnosis, significantly outperforming AI services. Error complementarity indicates potential for improvement through systematic integration of AI as a “second reader” rather than a primary diagnostic tool. However, the high false-positive rate of standalone AI requires substantial algorithm refinement. The optimal implementation strategy involves using AI as an auxiliary tool within radiologist workflows rather than as an autonomous diagnostic system, with potential for delayed verification protocols to maximize diagnostic sensitivity while managing the false-positive burden.

## 1. Introduction

Intracranial hemorrhages (ICHs) represent one of the most critical emergency conditions in neurology, requiring immediate diagnosis and treatment decision making. According to World Health Organization data, 3.4 million new ICH cases are registered annually worldwide [1], with mortality reaching 40% within the first month and 54% within a year [2]. The economic burden of stroke, including hemorrhagic forms, is estimated at USD 890 billion annually, comprising 0.66% of the global GDP [3,4].

Timely ICH diagnosis is critically important for patient survival and functional outcomes. The «golden hour» concept emphasizes that most hematoma volume expansion occurs within the first 6–24 h after hemorrhage [5]. However, despite brain computed tomography being considered the gold standard for ICH diagnosis [6], allowing for hemorrhage visualization with high sensitivity and specificity, modern emergency neuroimaging systems face difficulties. The average time from CT prescription to final radiologist conclusion in emergency departments is 5.9 h (median 4.2 h), with average interpretation delays during off-hours reaching approximately 4 h [7]. Native brain CT is analyzed more rapidly on average (8 min 25 s for cases without pathology and 13–14 min for cases with detected ICH) [8]; however, CT studies may await interpretation in the radiologist’s queue for extended periods due to case accumulation.

Global shortage of radiology specialists is exacerbated by increasing workload: between 2009 and 2020, diagnostic work volume increased by 80%, while the number of radiologists grew by only 2.5% [9]. Professional burnout affects 44% to 65% of radiologists, potentially impacting diagnostic quality [10].

In recent years, artificial intelligence (AI) systems have been considered a promising solution for optimizing ICH diagnosis. A meta-analysis by Karamian & Seifi, including 73 studies, showed that machine learning algorithms achieve pooled sensitivity of 92% (95% CI 90–94%) and specificity of 94% (95% CI 92–95%) in ICH detection [11]. In a randomized clinical trial by Yun et al., AI assistant use significantly improved diagnostic accuracy of both certified radiologists (2.82% increase, *p* = 0.0025) and clinical physicians (3.15% increase, *p* = 0.0072) [12].

By 2024, the US Food and Drug Administration (FDA) had approved over 750 AI services for healthcare, with 76% of all medical AI software relating to radiology [13]. Commercial systems like Viz.ai ICH and Canon AUTOStroke Solution demonstrate 85–93% sensitivity and 93–99% specificity for ICH diagnosis on CT under controlled clinical conditions [14]. However, despite these impressive results, the real clinical effectiveness of AI services remains a subject of debate [15]. Studies conducted within the large Moscow experiment on innovative computer vision technology implementation revealed significant reduction in AI service diagnostic accuracy for chest X-rays and brain CT when transitioning from retrospective to prospective clinical validation [16,17]. For some areas of diagnostic radiology, there is discussion of autonomous sorting using AI: studies demonstrate its effectiveness in chest X-ray analysis (fully correct autonomous sorting was achieved in 99.95% of cases, with clinically significant discrepancies recorded in 0.05% of cases) [18].

Existing studies have several important limitations. First, most work is based on retrospective data without direct comparison of AI and radiologists on identical image sets [11]. Second, the question of diagnostic error characteristics is insufficiently studied: are AI and physician errors complementary or duplicative? Third, the clinical significance of AI false-positive results in ICH diagnosis remains unclear [19], as does the optimal scenario for applying such AI services in emergency diagnosis where interpretation time is critically important. In the systematic review by Mäenpää & Korja, the lack of studies with reliable external validation of AI models for emergency neuroimaging is emphasized [20].

The objective of this study was to conduct a direct comparative analysis of diagnostic effectiveness between three commercial AI systems with state registration as medical devices operating alone versus radiologists with AI assistance in detecting ICHs on a sample of brain CT studies collected in real-time clinical practice over at least 15 months.

## 2. Materials and Methods

### 2.1. Study Design

This prospective, multicenter comparative study was conducted from April 2022 to December 2024 as part of the Moscow experiment on innovative computer vision technology implementation for medical image analysis [21]. The study was approved by the independent ethics committee of the Moscow Radiological Society (protocol No. 2 dated 20 February 2020) and registered on clinicaltrials.org (NCT04489992).

The study design reflects real clinical practice with access to all diagnostic information for decision making. This is a concordance study between different diagnostic approaches under complex clinical evaluation conditions.

The primary endpoint was comparison of diagnostic concordance between three commercial AI services and radiologist conclusions (with AI results available) in detecting intracranial hemorrhages on native brain CT images. Secondary endpoints included analysis of clinically significant diagnostic discrepancy patterns consisting of pathology misses and assessment of radiologist–AI complementarity.

### 2.2. Study Participants

#### 2.2.1. Inclusion and Exclusion Criteria

Inclusion criteria included the following:Patients over 18 years who underwent native brain CT in inpatient medical organizations;CT studies performed for clinical indications (suspected ICH, head trauma, acute neurological symptoms, and control after neurosurgical interventions);Technically adequate DICOM format images;Presence of primary radiologist conclusion.

Exclusion criteria included the following:Contrast-enhanced CT studies;Technically inadequate images (artifacts making analysis impossible and incomplete brain coverage);Absence of primary radiologist conclusion;Technical errors in in AI service analysis (absence of additional DICOM series, absence of accompanying information in DICOM SR standard, modification of original CT series, brightness/contrast of additional series not matching original image, and markup outside target organ).

#### 2.2.2. Sample Size

Sample size calculation was based on required confidence interval precision for main diagnostic metrics. To achieve ±2% precision for sensitivity and specificity at 95% confidence interval with expected sensitivity of 90%, we used Formula (1):n = Z^2^α/2 × p(1 − p)/d^2^(1)
where Z^2^α/2 = 3.84, p = 0.9, and d = 0.02.

The minimum required number of ICH cases was 864. With an expected ICH prevalence of 30%, the total sample size was determined as 2880 studies. Actual analysis was conducted on a sample of 3409 CT studies randomly selected from 439,542 studies processed by AI systems during the observation period. Of these, 1101 (32.3%) contained ICH signs; 2308 (67.7%) were without pathology. This sample size adequacy for binary classification tasks corresponds to published recommendations [22].

### 2.3. Study Setting

The study was conducted across 67 inpatient medical organizations in Moscow connected to the Unified Radiological Information Service of Unified Medical Information and Analytical Service (URIS UMIAS). CT studies were performed on 176 computed tomographs from four manufacturers: Siemens (Berlin, Germany), GE Healthcare (Chicago, IL, USA), Philips (Amsterdam, The Netherlands), Canon (Tokyo, Japan). All studies included series with the following parameters: window width = 80–100 HU for visualizing brain tissue density differences; window center = 30–50 HU for optimal white and gray matter differentiation; slice thickness = 0.5 mm; and data format = DICOM. All studies were non-contrast brain CTs performed for clinical indications. Accompanying clinical information included unique study identification number, patient age and sex, examination date, medical institution, primary description, and radiologist conclusion.

### 2.4. AI Services

The present study analyzed only those AI services that applied for participation in the Moscow experiment [21] and successfully completed all stages of validation and testing, including technical integration and clinical verification, which enabled their approval for real-world clinical practice within the Moscow healthcare system [23].

**AI Service 1 (AI-1)**: CELSUS^®^, developer LLC «Medical Screening Systems» (registration certificate 2022/18855). The architecture includes deep neural networks specifically adapted for medical image analysis. Its purpose is automatic detection of intracranial hemorrhages. The system automatically extracts data from PACS, analyzes images slice by slice, and returns two additional series: original images with pathology markup and visualization with quantitative parameters (volume, localization) in DICOM format (Figure 1), plus a structured report with preliminary description and numerical probability value in DICOM SR format. The average processing time is 30–60 s.

**AI Service 2 (AI-2)**: NTechMed CT Brain, developer LLC «NTech lab» (registration certificate 2024/22142). The architecture consists of neural network model complex for pathology detection, classification, and localization. Its purpose is diagnosis of hemorrhagic and ischemic strokes, including quantitative assessment and ASPECTS scale analysis. The technical architecture includes an integration bus (PACS-BUS) and AI server (AI Service). Models are grouped by separate tasks: brain CT detection, hemorrhagic area classification, and lesion localization (Figure 2). The processing time is up to 6 min.

**AI Service 3 (AI-3)**: «Third Opinion. CT Brain» software module of the Third Opinion Platform, developer Third Opinion Platform LLC (registration certificate 2024/23268). The architecture represents a CAD system based on supervised machine learning. Its purpose is searching for semiotic pathology signs with prioritization (triage) function. Functional capabilities include diagnosis of all ICH types (intracerebral, intraventricular, subarachnoid, subdural, and epidural), ischemic stroke detection with ASPECTS scale assessment, cystic-gliosis transformation detection, and automatic critical case prioritization (Figure 3). The processing time is up to 5 min.

### 2.5. Reference Standard

All CT studies were interpreted by radiologists within routine clinical practice. Conclusions were issued according to local protocols.

For reference standard creation, independent expert evaluation of a random sample of studies (n = 3409) processed by AI services was conducted by two radiologists: one with 6 years of neuroradiology experience and one with >15 years of neuroradiology experience. The expert evaluation procedure included independent image review by each expert followed by classification as ICH+ and ICH−. Experts had access to complete clinical information for each patient, including the entire CT study history and primary conclusions from radiologists at participating medical centers. In cases of expert disagreement, consensus discussion was conducted, and when consensus could not be reached, the study was excluded from analysis. Expert evaluation of AI service results and primary radiologist reports was performed in parallel as part of creating a unified reference standard. No separate quantitative comparative analysis between expert diagnoses and radiologist conclusions from individual medical centers was conducted.

ICH diagnostic criteria included the following:Hyperdense foci in brain parenchyma, meningeal spaces, or ventricular system (epidural, subdural, subarachnoid, and intracerebral localization);Density > 40 HU for acute hemorrhages;Exclusion of calcified areas;Minimum registered hemorrhage volume: 1 mm^3^.

The expert evaluation sample was collected according to developed methodology for AI service testing and clinical monitoring [23,24]: monthly analysis of 80 randomly selected CT studies. AI services were evaluated by two main criteria.

Interpretation (conclusion) correspondence was as follows:Complete correspondence: AI conclusion fully matches expert opinion (both for normal cases and pathology presence);Partially correct assessment: pathology presence confirmation with description disagreements (e.g., intracerebral interpreted as subarachnoid hemorrhage, etc.);False-positive result: AI indicated pathology rejected by expert;False-negative result: AI missed pathology confirmed by expert.

Localization (markup) correspondence included the following:Complete correspondence: accurate pathological zone localization;Partially correct assessment: correct detection with inaccurate localization (e.g., part of hemorrhage not marked);False-positive result: AI contoured a structure as pathology not confirmed by expert;False-negative result: AI did not contour pathology confirmed by expert.

All AI service diagnostic errors were documented with brief descriptions and screenshots. Conclusions about typical cases of clinically significant AI service errors consisting of ICH misses were formed. In parallel, radiologist errors were counted and analyzed: ICH presence was compared according to expert markup with the actual ICH description in radiologist conclusions.

### 2.6. Workflow and Technical Integration

The clinical workflow included several stages:(1)CT study performed in medical institution for clinical indications;(2)Images automatically transmitted to URIS UMIAS;(3)Active AI services processed CT study (processing time did not affect clinical process but took no more than 6 min);(4)Radiologist formed conclusion with access to AI analysis results as auxiliary tool within usual clinical practice;(5)AI results and physician conclusions saved in database for subsequent analysis;(6)Monthly expert evaluation of random sample of 80 studies for each active AI service.

The implementation of AI services for brain CT hemorrhage diagnosis into routine clinical practice began in April 2022 as part of the Moscow Experiment. Radiologists had no prior access to such AI diagnostic technologies before this implementation.

Technical integration of AI services into URIS UMIAS (Figure 4) enabled automatic extraction of needed DICOM images from the system, end-to-end analysis from source data loading to final formation, and output of additional visual data as DICOM image series with markup along with native images, plus structured reporting documentation in DICOM SR standard. Results transmission was accomplished as follows: the Apache Kafka platform generated events, forming JSON objects with numerical values of pathological change detection probability. Data synchronization occurred through Kafka as a connecting link between AI services and URIS UMIAS.

AI services processed CT studies independently of each other. Radiologists had access to AI service results within usual clinical practice and could use them as auxiliary tools when forming conclusions. Nevertheless, such utilization remained optional and was conducted at the discretion of individual practitioners. Experts conducting verification had access to primary radiologist conclusions and AI service results, reflecting real clinical practice of comprehensive diagnostic evaluation. The case presentation sequence to experts was randomized to minimize systematic errors.

### 2.7. Statistical Analysis

Statistical data processing was conducted using Python (version 3.8+) and a specially developed web tool for ROC curve analysis (https://roc-analysis.mosmed.ai/, accessed on 1 August 2025). Statistical significance level was set at 0.05 for all conducted tests.

Service performance quality was evaluated by the following classification metrics:Sensitivity (Se) = TP/(TP + FN) × 100%;Specificity (Sp) = TN/(TN + FP) × 100%;Overall accuracy (Ac) = (TP + TN)/(TP + TN + FP + FN) × 100%;Positive predictive value (PPV) = TP/(TP + FP) × 100%;Negative predictive value (NPV) = TN/(TN + FN) × 100%;Diagnostic odds ratio (DOR) = (TP × TN)/(FP × FN),
where TP = true-positive, TN = true-negative, FP = false-positive, and FN = false-negative results according to expert evaluation.

Metric confidence intervals were calculated using bootstrap (Python, version 3.8+), with 2.5 and 97.5 percentiles of bootstrapped samples taken as lower and upper bounds. For DOR, logarithmically transformed confidence intervals were used with subsequent inverse transformation.

#### 2.7.1. Comparative Analysis Between AI Services

The following methods were applied for diagnostic metric comparison between AI services: non-parametric Mann–Whitney test for comparing monthly metric values between services with null hypothesis formulation: «metric distributions in compared services are identical»; Pearson correlation coefficient for analyzing relationships between service operation duration and their diagnostic metrics. Cohen’s h effect size was calculated for quantitative assessment of proportion differences using the formula: h = 2 × (arcsin (√p_1_) − arcsin (√p_2_)).

#### 2.7.2. Comparative Analysis of AI Services and Radiologists

Comparison was conducted on an identical case sample (paired data). McNemar’s test was used for comparing sensitivity, specificity, and overall accuracy with χ^2^ calculation using the formula χ^2^ = (|b − c| − 1)^2^/(b + c), where b and c are numbers of discordant pairs. For PPV, NPV, and DOR comparison, analysis of non-overlapping 95% confidence intervals was applied, with the absence of interval overlap interpreted as a statistically significant difference (*p* < 0.05). For quantitative clinical significance assessment, Cohen’s h effect size of differences was calculated with interpretation: 0.2 = small effect, 0.5 = medium effect, 0.8 = large effect, and >1.0 = very large effect.

Agreement between AI services and radiologists was evaluated using Cohen’s kappa coefficient (κ) calculated by the formula κ = (P_0_ − P_e_)/(1 − P_e_), where P_0_ = observed agreement, and P_e_ = expected random agreement, with interpretation; κ < 0.20 = poor agreement, 0.21 − 0.40 = fair, 0.41 − 0.60 = moderate, 0.61 − 0.80 = substantial, and 0.81 − 1.00 = almost perfect agreement. Detailed analysis of agreement and disagreement patterns was conducted using 2 × 2 contingency tables.

All statistical tests were two-sided. Multiple comparison correction was not applied.

## 3. Results

### 3.1. Comparison of AI Services Among Themselves

During April 2022 to December 2024, three AI services (AI-1, AI-2, and AI-3) designed for diagnosing intracranial hemorrhages from computed tomography data were integrated into 67 medical organizations, providing data analysis from 176 diagnostic devices. During this period, systems processed 439,542 brain CT studies: 67,704 in 2022, 144,896 in 2023, and 226,942 in 2024. Expert verification (Ground Truth, GT) was conducted for 3409 CT studies: AI-1-1200 cases (580 or 48.3% with pathology signs), AI-2-1138 cases (238 or 20.9% with pathology signs), and AI-3-1071 cases (283 or 26.4% with pathology signs).

The patient cohort was characterized by a mean age of 64.3 ± 15.6 years, with a 46% female proportion. Binary classification results demonstrated variability in diagnostic effectiveness across services. Details of sensitivity, specificity, accuracy, and AUROC indicators are presented in Table 1.

Sensitivity shows how well the AI service detects true pathology cases. AI-1 demonstrated the highest sensitivity; however, this difference was not statistically significant compared to AI-2 and AI-3, whose differences were also not statistically significant. All services demonstrate equally high AUROC values, indicating their good diagnostic capabilities.

However, AI-2 and AI-3 demonstrated statistically significantly higher specificity compared to AI-1 (*p*-value < 0.001), indicating fewer false-positive activations. AI-2 and AI-3 also had similar accuracy that was statistically significantly higher (*p*-value = 0.004 and 0.002, respectively) than AI-1.

Correlation analysis revealed differences in the relationship between service operation duration and their diagnostic metrics. For AI-1, statistically significant moderate positive correlation was observed between operation time and specificity as well as accuracy, with Pearson correlation coefficients of 0.5 at *p*-value = 0.04. Additionally, moderate positive correlation was found for AUROC (Pearson’s rho = 0.6, *p*-value = 0.03). This indicates that with increased operation time, AI-1′s ability to correctly identify cases without pathology and overall diagnostic accuracy improve. No statistically significant relationship between sensitivity and service operation duration was found.

Unlike AI-1, correlation analysis for AI-2 and AI-3 revealed no statistically significant relationship between operation duration and diagnostic metrics. This indicates their effectiveness remained stable over time and did not depend on usage duration.

### 3.2. Comparison of AI and Radiologists

Parallel to AI service evaluation, comparative analysis of diagnostic accuracy between radiologists and AI services was conducted on an identical CT study sample. The comparison was based on radiologist conclusions with access to AI analysis results as auxiliary tools, reflecting real clinical practice of such technology implementation. Details are presented in Table 2.

Radiologists demonstrated statistically significant superiority across all main diagnostic metrics. Sensitivity was 98.91% vs. 95.91% for AI, specificity was 99.83% vs. 87.35%, and overall accuracy was 99.53% vs. 90.11%. Among AI services, AI-3 showed the highest accuracy as a combination of sensitivity and specificity (92.6%).

The radiologists’ diagnostic odds ratio exceeded that of AI by 323-fold (52,272 vs. 162.0), indicating a qualitatively different level of diagnostic capability. The greatest differences were found in specificity (Cohen’s h = 0.64, large effect size) and positive predictive value (Cohen’s h = 0.85, large effect).

AI services missed 45 ICH cases (4.09%) vs. 12 cases for radiologists (1.09%). The critical difference was the false-positive rate: 293 cases for AI (12.65%) vs. 4 cases for radiologists (0.17%), a 73-fold increase. The clinical burden from false-positive results differs substantially: with AI use, 1 false-positive result is expected per 9 studies vs. 1 per 577 studies for radiologists.

Agreement analysis between AI services and radiologists showed substantial agreement (Cohen’s kappa 0.776). Overall agreement was 89.6%, with the contingency table demonstrating the following distribution:1044 cases (30.6%): agreement on positive result;49 cases (1.4%): radiologist “+” and AI “−”;305 cases (8.9%): radiologist “−” and AI “+”;2011 cases (59.1%): agreement on negative result.

The independence of ICH misses is noteworthy: missed case sets did not overlap. All 12 cases undetected by radiologists were found by AI services (Table 3), and all 45 cases missed by AI were diagnosed by radiologists. This complementarity indicates different mechanisms of diagnostic error occurrence.

Detailed analysis of 12 CT studies with intracranial hemorrhages missed by radiologists demonstrated that in all these cases, hemorrhages were identified by AI services (Figure 5). Conversely, in all 45 CT studies where AI services did not detect hemorrhages, pathology was identified by radiologists. Most false-negative results from AI services concerned subarachnoid hemorrhages (75%) and small hemorrhages located near bone structures, as well as in the tentorial and convexital regions (83%).

Analysis of combined method potential revealed high potential for diagnostic error reduction. In the studied sample, ICH miss overlap was absent, which, in an ideal scenario, would allow for the elimination of all diagnostic errors. However, statistical uncertainty assessment shows that with a small error sample (12 radiologist misses), the 95% confidence interval for overlap probability is 0–26.5%. This means the possibility of 0 to 3 common misses under similar conditions, corresponding to realistic combined sensitivity of 99.7–100.0%. A potential limitation of the combined approach is substantial increase in false-positive results (from 4 to 297 cases) due to AI service error summation, requiring additional specialist verification and increased workload.

## 4. Discussion

When comparing AI services in routine clinical flow, AI-1 demonstrated highest sensitivity; however, its specificity and accuracy were significantly lower than other services, indicating more false-positive results. AI-2 and AI-3 showed statistically significantly superior but similar results to AI-1 in accuracy and specificity, making them more reliable but not ideal in excluding false-positive diagnoses.

It should be noted that AI services designed for intracranial hemorrhage detection demonstrate stable diagnostic capabilities when evaluating pathology within binary classification according to recent data. Key studies conducted in real clinical conditions show AI services achieve sensitivity ranging from 92% to 98.1% and specificity from 89.7% to 96% compared to expert consensus standards [25,26,27,28]. Balanced metric profiles are noted as superior. For example, VeriScout™ AI service metrics (96% specificity and 92% sensitivity) create optimal conditions for work list prioritization, maintaining high true-positive proportion without overloading radiologists with false alerts [29]. Diagnostic metrics obtained from our analysis of three AI services fall within described ranges, indicating their maturity and comparably high diagnostic value, even despite artifacts and surgical intervention consequences present in clinical flow, demonstrating robustness to common «constraining» factors.

High negative predictive value (NPV) is noted as a critically important advantage in publications, reflecting the proportion of true-negative cases among all cases the AI service determined as negative. Algorithms integrated into scanners can achieve 99.7% NPV, effectively excluding hemorrhages, which is particularly valuable in overloaded emergency departments where rapid triage is important [25]. However, positive predictive value (PPV) variability—the proportion of true-positive cases among all cases the service determined as positive—remains problematic, fluctuating from 56.7% in multicenter validation conditions to 98.4% in single-center studies and model optimization, reflecting differences in training data composition and clinical deployment conditions [30,31].

AI-1 in our study has high sensitivity, meaning it detects true pathology cases well, but its specificity is lower than other services. This explains the moderate PPV (78.7%) and high NPV (96.9%). AI-2 and AI-3 have more balanced sensitivity and specificity indicators, leading to higher PPV and NPV. Such AI services are suitable for general radiological flows requiring simultaneous CT study prioritization without excessive diagnostic dependence, while having high sensitivity and NPV is higher priority for emergency triage, where rapid ICH exclusion affects resource allocation and patient flow. However, specificity becomes the determining limiting factor.

Our study results convincingly demonstrate persistent statistically significant superiority of radiologists over AI services across all main diagnostic metrics. Achieved radiologist sensitivity of 98.91% and specificity of 99.83% significantly exceed indicators presented in most modern AI service studies for intracranial hemorrhage diagnosis [8,32,33,34,35,36]. Radiologists’ diagnostic odds ratio exceeding AI services by 323 times reflects a qualitatively different level of diagnostic capability. This difference is particularly important in emergency neuroimaging context, where diagnostic accuracy directly affects clinical outcomes and healthcare resource allocation. Some studies report that AI assistance significantly improves radiologist diagnostic accuracy [12]; however, in real clinical conditions similar to those in our study, no significant differences in accuracy between radiologists with and without AI assistant (99.2% vs. 99.5%) were established [15]. Protocol preparation time also did not improve (149.9 min vs. 147.1 min), raising questions about AI implementation effectiveness criteria in real clinical practice.

The critical limitation of AI services is the extremely high false-positive rate: 73 times higher than radiologists. This problem was addressed in our previous study [17], and many authors report it, noting main causes as postoperative changes (23.6%), image artifacts (19.7%), and tumors (15.3%) [31,32,37]. This creates serious additional burden on radiology specialists. Such differences may negate potential time savings from AI use and create «diagnostic noise», reducing specialist confidence in technology, and even lead to «alert fatigue» [38] and burnout [39], representing serious clinical implementation problems.

Analysis of intracranial hemorrhage characteristics missed by radiologists revealed patterns consistent with diagnostic error factors described in the literature. Subarachnoid hemorrhages were most frequently missed (33% of cases, 4/12), which corresponds to reported substantial rates of false-negative CT interpretations in aneurysmal SAH even by experienced neuroradiologists [40]. The predominance of misses in anatomically complex regions (tentorial 25% and parafalcine spaces 17%), as well as small-volume hemorrhage misses (75%), can be explained by both technical limitations of CT imaging and cognitive perceptual factors caused by fatigue and high clinical workload on specialists in emergency care settings. In our study, radiologists had access to AI results but, for unknown reasons, did not utilize them, leading to false-negative interpretations. Regarding AI service hemorrhage misses, our data are also consistent with findings presented in the literature. A prospective study of clinical AI service implementation for ICH detection demonstrated that small subdural hematomas (69.2%) and SAH (77.4%) represent the greatest diagnostic challenges [32].

The discovered complete complementarity of ICH misses in this sample represents an important finding with fundamental significance for AI clinical implementation strategies. The absence of overlap between cases missed by radiologists and AI services indicates different diagnostic error occurrence mechanisms and creates theoretical potential for their near-complete elimination with combined reading. Chen et al. in a systematic review report three main radiologist–AI collaboration models: parallel AI reading of diagnostic images, using AI as second reader, and AI as screening/triage all lead to diagnostic image analysis time reduction [41]. However, agreement indicator analysis with Cohen’s kappa 0.776 indicates substantial but not ideal agreement between experts and AI, which may partly be explained by different approaches to emergency CT interpretation and AI prediction underestimation [42].

Medical institutions implementing such solutions should consider local factors: emergency department load, radiologist availability, and baseline missed pathology levels. For example, in trauma centers, AI services with high PPV may prove more useful despite computational costs, while in general hospitals, triage optimized for NPV may be prioritized [43]. Systems achieving >95% sensitivity with >90% specificity demonstrate sufficient reliability for auxiliary use, while AI services below these thresholds risk being clinically impractical due to error frequency [44].

To minimize negative consequences, it is recommended to use AI services for intracranial hemorrhage diagnosis only as second reading tools with mandatory verification of all positive results by radiologists, possibly not immediately after AI CT study processing but at shift end or at certain time intervals to avoid distracting and burdening attention in the moment. It is also extremely important to improve AI service specificity to reduce false activations and train physicians to effectively filter AI conclusions, understanding their weaknesses and most probable error types and localizations.

## 5. Limitations

This study has several important methodological limitations that must be considered when interpreting results. First, the radiologists had access to AI analysis results within usual clinical practice, which could potentially improve their diagnostic performance compared to completely independent work. This reflects real AI technology implementation practice but limits possibilities for evaluating “pure” AI effect on diagnostic accuracy. Second, reference standard creation was conducted by experts with access to both primary radiologist conclusions and AI service results. Third, analysis was conducted only on CT studies successfully processed by all AI services. Cases with technical analysis errors were excluded, which could lead to the overestimation of real AI service effectiveness in routine clinical practice conditions. Also, different AI services operated during different time periods with various durations, which could affect analyzed case composition and processing conditions. Fourth, analysis did not account for factors such as case complexity, radiologist experience, study timing, study urgency, comorbid pathology presence, and some others, preventing the establishment of optimal AI service use conditions. Fifth, results were obtained under specific healthcare system conditions with certain organizational features and may not fully generalize to other medical systems with different work protocols and specialist training levels.

## 6. Conclusions

This prospective, multicenter study convincingly demonstrates statistically significant superiority of radiologists over AI services across all diagnostic metrics (*p* < 0.001): sensitivity 98.91% vs. 95.91% and specificity 99.83% vs. 87.35%, with diagnostic odds ratio exceeding AI by 323-fold. The critical AI limitation is a very high false-positive rate—73 times higher than radiologists (12.65% vs. 0.17%)—making autonomous application impossible. However, the key finding is complete complementarity of diagnostic errors: all 12 radiologist misses were detected by AI, while all 45 AI misses were diagnosed by radiologists, creating potential for 99.7–100% combined sensitivity. Thus, radiologists maintain their role as the “gold standard” in intracranial hemorrhage diagnosis on brain CT, and the optimal AI implementation strategy in this diagnostic situation involves using it as a “second opinion” tool with mandatory but possibly delayed verification of positive results by specialists to minimize diagnostic misses.

## Figures and Tables

**Figure 1 jcm-14-05700-f001:**
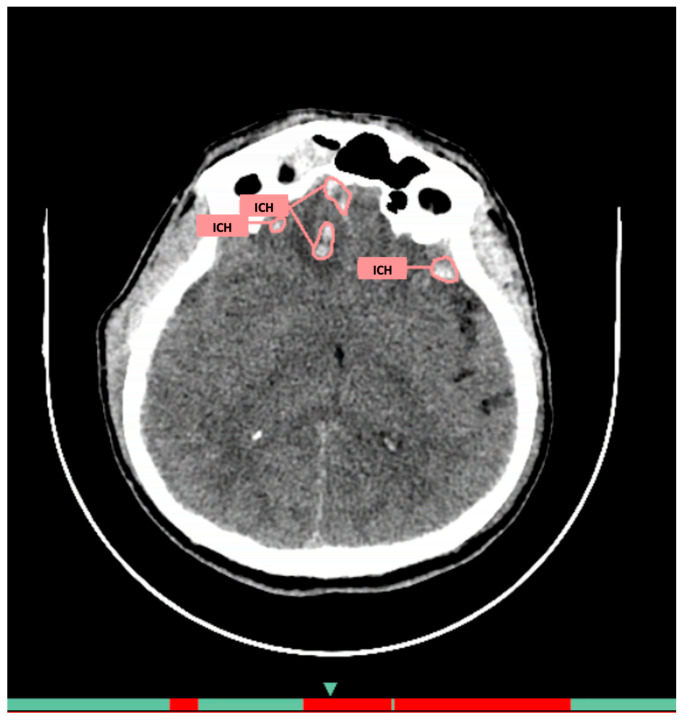
Example of functioning AI-1 for the diagnosis of intracranial hemorrhages on brain CT.

**Figure 2 jcm-14-05700-f002:**
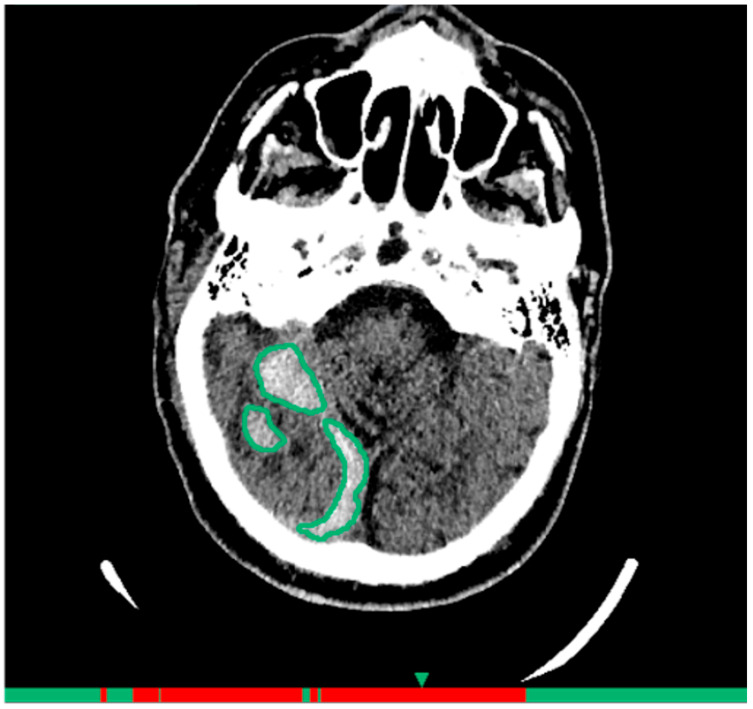
Example of functioning AI-2 for the diagnosis of intracranial hemorrhages on brain CT.

**Figure 3 jcm-14-05700-f003:**
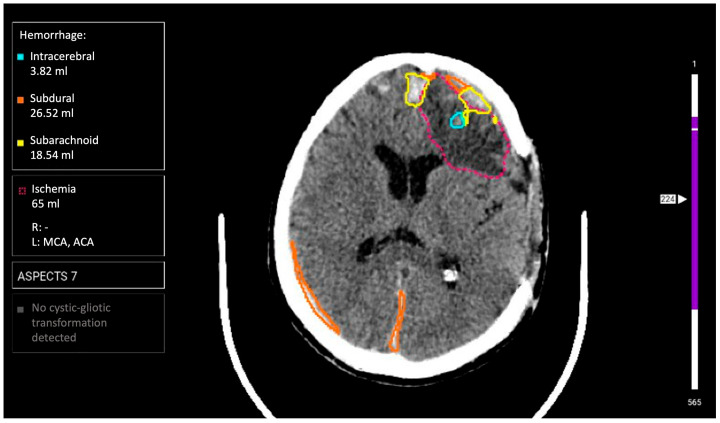
Example of functioning AI-3 for the diagnosis of intracranial hemorrhages on brain CT.

**Figure 4 jcm-14-05700-f004:**
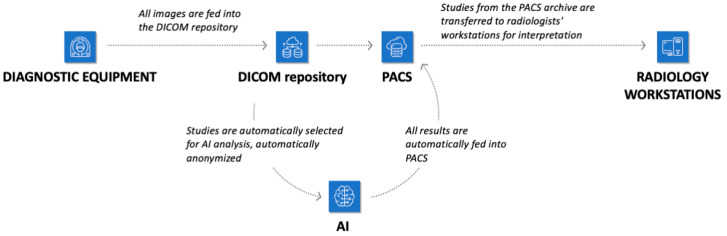
Infrastructure of the city’s radiology service with the possibility of connecting AI tools.

**Figure 5 jcm-14-05700-f005:**
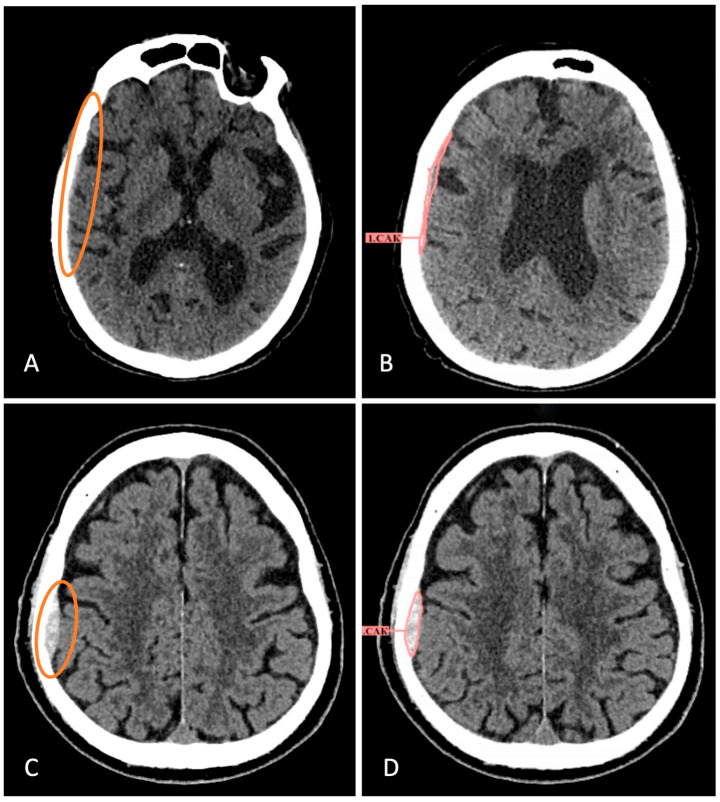
Examples of missed hemorrhages by radiologists that have been identified by AI services. (**A**) Brain CT scan with atrophic changes of the brain, microangioencephalopathy, total substitution hydrocephalus (from report). Small-volume SDH in the right temporoparietal region was not described. (**B**) AI service partially detected it, although the type was incorrectly indicated. (**C**) The radiologist did not report EDH in the right parietal region after the patient’s fall. (**D**) The AI service successfully identified and segmented the hemorrhage. The hemorrhages that the radiologist didn’t notice are circled in color.

**Table 1 jcm-14-05700-t001:** AI service metrics calculated based on 15+ months of operation in routine clinical practice.

Characteristics	AI-1	AI-2	AI-3
Total CT studies (abs.)	1200	1138	1071
CT studies with pathology (GT) (abs.)	580	238	283
TP (abs.)	565	227	264
TN (abs.)	469	826	728
FP (abs.)	153	80	60
FN (abs.)	15	11	19
Sensitivity, % (95% CI)	97.4 (95.8–98.5)	95.4 (92.7–98.0)	93.2 (90.1–95.9)
Specificity, % (95% CI)	75.4 (71.8–78.7) *	91.2 (89.3–93.0)	92.3 (90.4–94.3)
Accuracy, % (95% CI)	86.0 (83.9–87.9) *	92.1 (90.5–93.6)	92.6 (90.1–94.1)
AUROC, % (95% CI)	92.6 (86.3–98.8)	93.5 (91.3–95.5)	91.7 (89.2–94.1)
PPV, % (95% CI)	78.7 (73.1–83.5)	68.5 (63.5–84.1) *	81.5 (78.6–91.2)
NPV, % (95% CI)	96.9 (95.2–99.1)	98.8 (97.9–99.9)	97.5 (95.7–99.3)

* statistically significant difference, *p*-value < 0.001.

**Table 2 jcm-14-05700-t002:** Comparative diagnostic characteristics of AI services and radiologists on identical brain CT study sample.

Characteristic	AI Services (Combined Data)	Radiologists	*p*-Value	Cohen’s h
General sample characteristics				
Total CT studies (abs.)	3409	3409	-	-
ICH+ (abs.)	1101	1101	-	-
ICH− (abs.)	2308	2308	-	-
Pathology proportion (%)	32.3	32.3	-	-
Main diagnostic metrics				
Sensitivity, % (95% CI)	95.91 (94.6–96.9)	98.91 (98.1–99.4)	<0.0001 ^†^	0.20
Specificity, % (95% CI)	87.35 (85.9–88.7)	99.83 (99.6–99.9)	<0.0001 ^†^	0.64
Accuracy, % (95% CI)	90.11 (89.1–91.1)	99.53 (99.3–99.7)	<0.0001 ^†^	0.50
PPV, % (95% CI)	78.28 (75.7–80.7)	99.63 (99.2–99.9)	<0.05 ^‡^	0.85
NPV, % (95% CI)	97.82 (97.2–98.3)	99.48 (99.1–99.7)	<0.05 ^‡^	0.15
DOR (95% CI)	162.0 (118.4–221.3)	52,272 (16,820–162,448)	<0.05 ^‡^	-
Diagnostic error analysis				
False negatives, abs. (%)	45 (4.09)	12 (1.09)	<0.0001 ^†^	-
False positives, abs. (%)	293 (12.65)	4 (0.17)	<0.0001 ^†^	-
Error overlap	0		-	-
Miss complementarity	Complete		-	-
Combined approach potential				
Observed scenario ^1^	-	100.0% sensitivity	-	-
Realistic estimate ^2^	-	99.7–100.0% sensitivity	-	-
Miss reduction ^3^	-	12→0–3 cases	-	-
Expected specificity ^4^	-	83.9%	-	-
Agreement indicators				
Cohen’s kappa (95% CI)	0.776 (0.750–0.801)	-	-	-
Overall agreement, %	89.6	-	-	-
ICH+ agreement, %	74.7	-	-	-
ICH− agreement, %	85.0	-	-	-

^†^ McNemar’s test for paired data (χ^2^ = 17.96 for sensitivity, χ^2^ = 279.27 for specificity, and χ^2^ = 291.08 for accuracy). ^‡^ Analysis of non-overlapping 95% Wilson confidence intervals for PPV, NPV, and DOR. ^1^ Scenario with complete absence of error overlap (observed in this sample). ^2^ 95% confidence interval accounting for statistical uncertainty (possible 0–3 common misses). ^3^ Range of physician miss reduction depending on error overlap degree. ^4^ Expected specificity when summing false-positive results (4 + 368 = 372 out of 2308).

**Table 3 jcm-14-05700-t003:** Characteristics of intracranial hemorrhages missed by radiologists but detected by AI services.

N	Age/Sex	ICH Type	Location	Volume/Characteristics	Clinical Context	Probable Miss Reason
1	79F	SAH	Multiple regions	Small volume	Hypertensive crisis, fall	Subtle SAH signs
2	21M	SAH	Parafalcine, tentorial	Small volume	Post-VPS surgery	Post-surgical changes
3	63M	SAH	Tentorial	Small volume	Head trauma	Minimal bleeding
4	71F	IPH	Parafalcine space	Small volume	Post-tumor resection	Surgical bed changes
5	67M	Chronic SDH	Right fronto-parieto-occipital	Mixed density, 40 HU	Stroke workup	Chronic appearance
6	45M	IPH	Right orbitofrontal	Small volume	Exclusion of acute intracranial pathology	Bone artifacts
7	86F	SDH	Right temporo-parietal	Small volume	Syncope, atrial fibrillation	Small size
8	81M	Chronic SDH	Bilateral	Mixed density, 39 HU	TIA symptoms	Chronic appearance
9	67M	IVH + SAH	Left temporal horn, tentorial	Residual blood	Post-drainage	Probably due to haste, the pattern of hemorrhages was not described accurately enough
10	90F	Hemorrhagic transformation	Ischemic zone	Petechial	Stroke progression	Ischemic mimicry
11	45M	EDH	Right parietal	Small volume	Alcohol-related fall, radial fracture	Trauma history
12	74F	Hemorrhagic transformation	Basal ganglia	Small volume	Hypertensive crisis	Early stroke phase

Abbreviations: SAH—subarachnoid hemorrhage; IPH—intraparenchymal hemorrhage; SDH—subdural hematoma; IVH—intraventricular hemorrhage; EDH—epidural hematoma; VPS—ventriculoperitoneal shunt; TIA—transient ischemic attack; HU—Hounsfield units.

## Data Availability

The raw data supporting the conclusions of this article will be made available by the authors on request.

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
