# Peer review of "Standalone AI Versus AI-Assisted Radiologists in Emergency ICH Detection: A Prospective, Multicenter Diagnostic Accuracy Study"

_jcm, 2025, doi:10.3390/jcm14165700_

Round 1
Reviewer 1 Report
Comments and Suggestions for Authors
In their current manuscript titled “Diagnostic Performance of AI versus Radiologists with AI in Emergency Brain CT: Intracranial Hemorrhage Detection Study” Khoruzhaya and colleagues present a timely and relevant prospective evaluation of AI performance compared to radiologists assisted by AI, conducted across multiple centers in Moscow. The study demonstrates a clear superiority of radiologists supported by AI over standalone AI, highlighting the potential for synergistic integration of AI tools in clinical neuroimaging workflows. There remain some comments to improve clarity of the results described.
It would be helpful to specify the study design (e.g., prospective, multicenter diagnostic accuracy study) within the title.
The manuscript does not clearly define the reference standard used for diagnostic confirmation. It appears to involve two expert neuroradiologists, but this should be explicitly stated. Furthermore, it would be important to clarify whether the gold standard diagnosis was compared with the individual center-level diagnoses.
The rationale for the choice of the AI tool used in this study is not provided. Was a specific tool selected based on prior validation, availability, or institutional preference?
Were the AI tools already in routine clinical use at the participating centers, or were they introduced solely for the purposes of this study? Additionally, were radiologists required to consult the AI output, or was its use discretionary?
The manuscript would benefit from a more detailed characterization of the hemorrhages missed by AI or radiologists, including type, size, and location.
Do the authors have some ideas about why some hemorrhages detected by AI were not identified by the radiologists?
Author Response
1. It would be helpful to specify the study design (e.g., prospective, multicenter diagnostic accuracy study) within the title.
Thank you for this helpful comment. We have discussed your suggestion and decided to modify the title to best reflect the essence of the study.
2. The manuscript does not clearly define the reference standard used for diagnostic confirmation. It appears to involve two expert neuroradiologists, but this should be explicitly stated. Furthermore, it would be important to clarify whether the gold standard diagnosis was compared with the individual center-level diagnoses.
We are grateful for this important clarification. We have revised Section 2.5 to explicitly state that our reference standard was created by two expert neuroradiologists (6 years and >15 years of neuroradiology experience) who had access to complete clinical information, entire CT study history, and primary conclusions from radiologists at all participating centers. We have also clarified that "no separate quantitative comparative analysis between expert diagnoses and radiologist conclusions from individual medical centers was conducted." Our reference standard design focused on comprehensive diagnostic evaluation using all available information to create a unified gold standard for comparing diagnostic methods (standalone AI vs. AI-assisted radiologists) rather than evaluating individual center performance.
3. The rationale for the choice of the AI tool used in this study is not provided. Was a specific tool selected based on prior validation, availability, or institutional preference?
The AI services included in our study were not selected by the research team based on prior validation or institutional preference. Rather, these were the only AI services that: 1) applied for participation in the official Moscow Experiment on AI implementation, 2) successfully completed mandatory technical integration testing, regulatory approval processes, and clinical validation phases, and 3) received official authorization for deployment in routine clinical practice across Moscow healthcare facilities. We have added clarifying text to Section 2.4 to explain this.
4. Were the AI tools already in routine clinical use at the participating centers, or were they introduced solely for the purposes of this study? Additionally, were radiologists required to consult the AI output, or was its use discretionary?
The AI tools were introduced specifically as part of the Moscow Experiment beginning in April 2022, and were not previously available in routine clinical practice at participating centers. Prior to this implementation, radiologists had no access to such AI diagnostic services. Regarding AI output consultation, radiologists had voluntary access to AI results within their standard clinical practice and could utilize them as auxiliary diagnostic tools when formulating conclusions, but this was discretionary rather than mandatory. We have added this clarification to Section 2.6 of the revised manuscript.
5. The manuscript would benefit from a more detailed characterization of the hemorrhages missed by AI or radiologists, including type, size, and location.
6. Do the authors have some ideas about why some hemorrhages detected by AI were not identified by the radiologists?
We are grateful for this valuable suggestion. We have added comprehensive characterization of missed hemorrhages as requested. Added content: 1) detailed table (Table 3) characterizing all 12 ICH cases missed by radiologists but detected by AI, including type, location, volume/characteristics, clinical context, and probable miss reasons; 2) discussion section analysis with probable explanation of miss factors, including technical CT limitations, cognitive perceptual factors due to fatigue and high clinical workload, and the paradoxical finding that radiologists had access to AI results but did not utilize them in these specific cases. The analysis demonstrates that AI detected hemorrhages typically missed due to small size, complex anatomical locations, post-surgical changes, and chronic appearance - areas where machine learning algorithms may have advantages over human pattern recognition under time pressure.
All additions are highlighted in yellow throughout the manuscript.
Reviewer 2 Report
Comments and Suggestions for Authors
I appreciate the authors for presenting this novel research article. which investigated the diagnostic effectiveness between AI services alone and radiologists with AI assistance in detecting ICH on brain CT. I all agree the authors recommend that optimal implementation strategy involves using AI as an auxiliary tool within radiologist workflow rather than autonomous diagnostic system, with potential for delayed verification protocols to maximize diagnostic sensitivity while managing false-positive burden. Thus, radiologists maintain their role as "gold standard" in intracranial hemorrhage diagnosis on brain CT, and optimal AI implementation strategy in this diagnostic situation involves using it as "second opinion" tool with mandatory but possibly delayed verification of positive results by specialists to minimize diagnostic misses. I have no more comments.
Comments on the Quality of English LanguageThe English could be improved to more clearly express the research.
Author Response
We thank the reviewer for this important feedback. We have thoroughly revised the entire manuscript to improve English language quality, including grammar, sentence structure, and scientific terminology. The revised version has been carefully reviewed to ensure clarity and adherence to publication standards.
We believe these improvements significantly enhance the manuscript's readability while maintaining scientific accuracy.
Round 2
Reviewer 1 Report
Comments and Suggestions for Authors
The authors have amswered all the raised concerns.